# Assessing the basic knowledge and awareness of dengue fever prevention among migrant workers in Klang Valley, Malaysia

**Maryam N. Chaudhary**[1], **Voon-Ching Lim**[2,3‡], **Erwin Martinez Faller**[4,5‡], **Pramod Regmi**[5‡], **Nirmal Aryal**[5‡], **Siti Nursheena Mohd Zain**[6‡], **Adzzie Shazleen Azman**[1], **Norhidayu Sahimin**[7]*

1 School of Science, Monash University Malaysia, Subang Jaya, Selangor, Malaysia, 2 Department of Biology, Faculty of Science, Chulalongkorn University, Bangkok, Thailand, 3 National Primate Research Center of Thailand, Chulalongkorn University, Saraburi, Thailand, 4 Pharmacy Department, School of Allied Health Sciences, San Pedro College, Davao City, Davao del Sur, Philippines, 5 Faculty of Health and Social Sciences, Bournemouth University, Bournemouth, United Kingdom, 6 Institute of Biological Sciences, Faculty of Science, Universiti Malaya, Kuala Lumpur, Malaysia, 7 Tropical Infectious Diseases Research and Education Centre (TIDREC), Universiti Malaya, Kuala Lumpur, Malaysia

☯ These authors contributed equally to this work.
‡ VCL, EMF, PR, NA and SNMZ also contributed equally to this work.
* ayusahimin@um.edu.my

## Abstract

### Background

Globally, 390 million dengue virus infections occur per year. In Malaysia, migrant workers are particularly vulnerable to dengue fever (DF) due to mosquito breeding sites exposure and poor health literacy. Therefore, this study aimed to (i) assess the current DF knowledge, attitudes and practices (KAP), and (ii) identify strategies to promote DF awareness, among migrant workers in Klang Valley.

### Method

A survey was conducted with 403 Nepali, Filipino and Indonesian migrant workers through phone interviews and online self-administered questionnaires. Piecewise structural equation modelling was applied to identify predictor variables for DF KAP.

### Results

Most respondents were male, working in the services industry, had completed high school, aged between 30–39 years and with less than ten years work experience in Malaysia. Overall, respondents' knowledge was positively correlated with attitude but negatively with practices. Older respondents, who had completed higher education, obtained higher knowledge scores. Similarly, those with working experience of >20 years in Malaysia obtained higher attitude scores. Respondents with a previous history of DF strongly considered the removal of mosquito breeding sites as their own responsibility, hence tended to frequently practise DF preventive measures. Respondents' knowledge was also positively correlated to their understanding of DF information sourced from social media platforms.

**Data Availability Statement:** All relevant data are within the manuscript and its Supporting Information files.

**Funding:** Ministry of Higher Education, Malaysia for niche area research under the Higher Institution Centre of Excellence (HICoE) program (MO002-2019 and TIDREC-2023) and the Malaysian Society of Parasitology and Tropical Medicine (MSPTM) for the Community Fund (PV008-2020) awarded to NS. The funders had no role in study design, data collection and analysis, decision to publish, or preparation of the manuscript.

**Competing interests:** The authors have declared that no competing interests exist.

## Conclusion

These findings highlighted: (i) the need for targeted DF educational intervention among younger and newly arrived workers with lower levels of education and (ii) maximising the usage of social media platforms to improve DF public awareness.

## Introduction

Dengue fever (DF) is the world's most rapidly spreading mosquito-borne viral disease, with no cure to-date [1, 2]. The causal agent of DF is a virus belonging to the Flaviviridae family, which is transmitted through female *Aedes aegypti* mosquitoes [1, 2]. DF can be fatal, with symptoms including high fever, body pains and persistent vomiting [3]. Globally, around 390 million dengue virus infections occur every year, with 70% of reported cases occurring in Asia [4]. In 2019, the highest number of dengue cases worldwide were reported, with 131,000 cases in Malaysia alone [3]. In 2020, an incidence rate per 100,000 population of 276.11 was recorded for DF in Malaysia [5]. However, national surveillance systems often underestimate the total number of symptomatic DF cases due to false negatives in diagnoses, inaccurate case-reporting from hospitals and low surveillance budget [6]. High rural-urban migration combined with poverty, often results in poor living standards such as poorly built homes with drainage issues, water supply shortages and poor waste management. These factors are ideal for the creation of breeding sites for the dengue vector [7].

In Malaysia, there is a high demand for low skilled labour, bringing large numbers of migrant workers for temporary employment [8]. Many live in overcrowded dormitories or informal settings near their workplace with inadequate water supplies which may compel them to store water in containers and inadvertently house mosquito breeding sites [9]. Most migrant workers in Malaysia are employed in the manufacturing (36%) and construction industry (19%) [10], exposing them to DF as outdoor construction sites are common mosquito breeding sites due to stagnant surface water [11]. Migrant workers often lack access to health-related information, therefore exhibiting poor health literacy, which results in their poor healthcare [12]. This may be attributed to language barriers, cultural differences and limited knowledge of their rights, as seen in Sweden [13], Thailand [14] and Italy [15]. Therefore, migrant workers are particularly vulnerable to DF, and may risk transmitting DF to others.

To identify preventative strategies for DF, numerous studies have evaluated the knowledge of, attitudes towards, and practices in, (KAP) DF prevention [16–20]. However, DF KAP among migrant workers in Malaysia remains understudied despite being a sizable portion of the country's workforce and vulnerable to DF [21]. Therefore, this study examined the DF KAP among migrant workers across occupational sectors in Klang Valley, Malaysia to (i) better understand the level of and relationship between knowledge, attitude and practice, (ii) identify specific cohorts within migrant workers to target for awareness campaigns, and (iii) examine the potential of social media as a platform for raising DF awareness.

## Methods

### Ethical consideration

This study was approved by the Universiti Malaya Research Ethics Committee (UM.TNC2/UMREC_1162). Informed consent was obtained from all participants.

## Questionnaire development

A preliminary questionnaire was first developed in English. A pilot study was conducted to test the comprehension of the questionnaire wherein five study leaders distributed the questionnaire to ~10 respondents each. The results of the pilot study were excluded from the main analysis.

The final questionnaire consisted of six parts (S1 File). Part A contained seven demographic questions about gender, age, nationality, level of education, district of residence, current occupation sector and date of commencing work in Malaysia. Part B contained five questions to elicit the illness history of the respondents. Part C examined the respondents' knowledge of DF and comprised eleven statements, each requiring respondents to choose 'True', 'False' or 'I am not sure' as their response. Part D explored the attitudes towards DF of respondents and contained six statements, each requiring respondents to choose 'Strongly agreed', 'Agreed', 'Neutral', 'Disagreed', or 'Strongly disagreed' as their response. Part E evaluated the DF prevention practices of respondents and contained seven statements, each requiring respondents to choose 'Usually', 'Sometimes' or 'Never' as their response. Part F assessed what sources respondents used to obtain DF information and their understanding of the information from these sources; it contained eight questions, each requiring respondents to choose either 'Yes' or 'No'.

## Data collection

Data collection occurred in Klang Valley, between mid-2020 and 2021, during the government-imposed Movement Control Order (MCO) in Malaysia as a response to the COVID-19 pandemic. The questionnaire was distributed using the Google Forms platform and was shared with the coordinators (representatives of the migrant worker communities) to distribute to respondents. Respondents from the Philippines and Nepal were invited for this study due to past collaboration with their respective coordinators, whereas Indonesian respondents were invited as they constitute the majority of migrant workers in Malaysia [22]. Filipino respondents completed the questionnaire in English. Due to limited English proficiency among Nepali and Indonesian respondents, Nepali respondents underwent a phone interview conducted by their coordinator in their native language, whereas Indonesian respondents completed the questionnaire in the Indonesian language. The Krejcie and Morgan formula was applied, with a 5% margin of error and 95% confidence interval, resulting in a minimum sample size of 385 participants [23]. A total of 455 individuals were invited for this study; 428 consented to completing the survey, of which 403 responses were recorded without erroneous/incomplete data and were used for data analyses.

## Data analyses

All data analyses [24] were conducted, using R version 4.2.1 [25]. The internal consistency of the KAP questions/statements in the questionnaire was assessed using Cronbach's Alpha test, using 'alpha()' from the 'psych' package [26]. A Cronbach's alpha value of $\geq 0.7$ indicates acceptable internal consistency of the questions to represent a single construct [27–29]. For Part C, an initial Cronbach's alpha of 0.52 was obtained. Upon removal of the statements: '*Only female mosquitoes suck bloods*', '*Mosquitoes lay their eggs in stagnant and dirty water*', '*Dengue fever can be cured only by taking paracetamol*', and '*Symptoms of dengue fever include fever, joint pain and rash*', the value improved to 0.73; therefore, the remaining seven statements were averaged to represent a construct C = 'Knowledge'. For Parts D and E, the Cronbach's alpha values calculated were 0.97 and 0.93, therefore, the scores for all statements were averaged to represent the constructs, D = 'Attitude', and E = 'Practice'. The statement '*Removal*

*of mosquito breeding sites is my responsibility*' was run as a separate construct, 'Responsibility', as it followed a trend opposite to the remaining Attitude statements.

Linear mixed-effects modelling [24, 30] was conducted using 'lmer()' from the 'lme4' package [31]. The explanatory variables were 'Gender', 'Age', 'Years Working in Malaysia', 'Education', 'Occupation', 'Dengue History', 'Understanding of DF Information from Social Media' and 'Responsibility', whereas the response variables were DF prevention 'Knowledge', 'Attitude' and 'Practice'. 'Nationality' was coded as a random error as the method of response collection varied between nationalities (e.g., English/Indonesian questionnaire, and phone interview). The categorical variables were converted to numeric binary (e.g. 0 = male and 1 = female) or ordinal (e.g., 1 = no formal education, 2 = primary school, 3 = high school, 4 = university). The final models were selected by removing predictors from a global model sequentially until all predictors in the model met the preselected criterion ($p < 0.05$) and all those outside did not. 'r.squaredGLMM()' was used to obtain the conditional and marginal $R^2$ [32], whereas 'ggplot()' from the 'ggplot2' package was used to visualise the data [33].

The selected models were used to create an overarching model via piecewise structural equation modelling (PSEM) [18, 19, 24, 34], using 'psem()' from the 'piecewiseSEM' package [35]. Based on the linear mixed-effects modelling results, 'Knowledge', 'Attitude' and 'Practice' were considered to be correlated rather than directly causally associated, and therefore were indicated as correlated errors in the PSEM. Selection of the final model was based on tests of directed separation ($p > 0.05$), which indicated that all variables were independent of one another, and Fisher's C test ($p > 0.05$), which confirmed that all potential paths were included in the model [35]. Using 'fitMeasures()' from the 'lavaan' package [36, 37], the comparative fit index (CFI), Tucker–Lewis index (TLI) and standardised root mean square residual (SRMR) were obtained to assess whether the PSEM fit the data well [18, 24, 34].

For post-hoc analyses, Wilcoxon rank-sum tests were performed using 'wilcox.test()', for binary data. For non-binary data, Kruskall-Wallis and Dunn's tests were conducted using 'kruskal.test()' and 'dunn_test()' from the 'rstatix' package [38].

## Results

Of the 403 respondents, the respondents were mostly Nepali, followed by Filipino and Indonesian. The majority were male, employed in the services industry, aged between 30–39 years, had completed high school education and with work experience in Malaysia of less than ten years (Table 1). Few respondents (2.0%, *n* = 8) stated that they had previously suffered from DF, and were either Indonesian (62.5%, *n* = 5) or Filipino (37.5%, *n* = 3). Respondents mostly obtained (Fig 1A) and understood (Fig 1B) information related to DF from social media (94.8%; 95.3%) and least from posters/billboards (56.8%; 61.3%).

### Dengue prevention KAP

The level of DF prevention knowledge among migrant workers was moderately high, with an average percentage score of 78.7±0.01 (Fig 2A). Nearly all respondents correctly answered the statements '*dengue fever is caused by mosquitoes*' (97.0%). Conversely, the statement that was least correctly answered was '*mosquitoes lay their eggs in stagnant and dirty water*' (3.2%). The respondents mostly showed positive attitudes towards DF prevention, with an average percentage score of 78.6±0.04 (Fig 2B). The statement with the highest combined agreement rate was '*dengue fever is very dangerous and can be fatal*' (75.9%), whereas the statement with the lowest agreement rate was '*removal of mosquito breeding sites is my responsibility*' (36.5%). On average, respondents usually/sometimes exercised appropriate DF preventative practices only 56.5±0.02 percent of the time. (Fig 2C). The most exercised practice was '*regular use of*

**Table 1. Socio-Demographic profile of the 403 respondents.**

| Variable | Nepali | | Indonesian | | Filipino | | Overall | |
|---|---|---|---|---|---|---|---|---|
| | n | % | n | % | n | % | n | % |
| **Sex** | | | | | | | | |
| Male | 201 | 96.2 | 22 | 23.9 | 20 | 19.6 | 243 | 60.3 |
| Female | 8 | 3.8 | 70 | 76.1 | 82 | 80.4 | 160 | 39.7 |
| **Education** | | | | | | | | |
| No formal education | 0 | 0 | 7 | 7.6 | 2 | 2.0 | 9 | 2.2 |
| Primary school | 89 | 42.6 | 26 | 28.3 | 2 | 2.0 | 117 | 29.0 |
| High school | 111 | 53.1 | 58 | 63.0 | 44 | 43.1 | 213 | 52.9 |
| University | 9 | 4.3 | 1 | 1.1 | 54 | 52.9 | 64 | 15.9 |
| **Age** | | | | | | | | |
| <20 | 0 | 0 | 6 | 6.5 | 0 | 0 | 6 | 1.5 |
| 20–29 | 38 | 18.2 | 44 | 47.8 | 4 | 3.9 | 86 | 21.3 |
| 30–39 | 146 | 69.9 | 25 | 27.2 | 40 | 39.2 | 211 | 52.3 |
| 40–49 | 25 | 12.0 | 14 | 15.2 | 41 | 40.2 | 80 | 19.9 |
| >49 | 0 | 0 | 3 | 3.3 | 17 | 16.7 | 20 | 5.0 |
| **Occupation** | | | | | | | | |
| Construction | 6 | 2.9 | 4 | 4.3 | 5 | 4.9 | 15 | 3.7 |
| Domestic helpers | 0 | 0 | 14 | 15.2 | 74 | 72.5 | 88 | 21.8 |
| Manufacturing | 42 | 20.1 | 0 | 0 | 10 | 9.8 | 52 | 12.9 |
| Plantation | 3 | 1.4 | 2 | 2.2 | 0 | 0 | 5 | 1.2 |
| Services | 158 | 75.6 | 72 | 78.3 | 13 | 12.7 | 243 | 60.3 |
| **Years Working in Malaysia** | | | | | | | | |
| <10 | 208 | 99.5 | 81 | 88.0 | 66 | 64.7 | 355 | 88.1 |
| 10–20 | 0 | 0 | 8 | 8.7 | 23 | 22.5 | 31 | 7.7 |
| >20 | 1 | 0.5 | 3 | 3.3 | 13 | 12.7 | 17 | 4.2 |

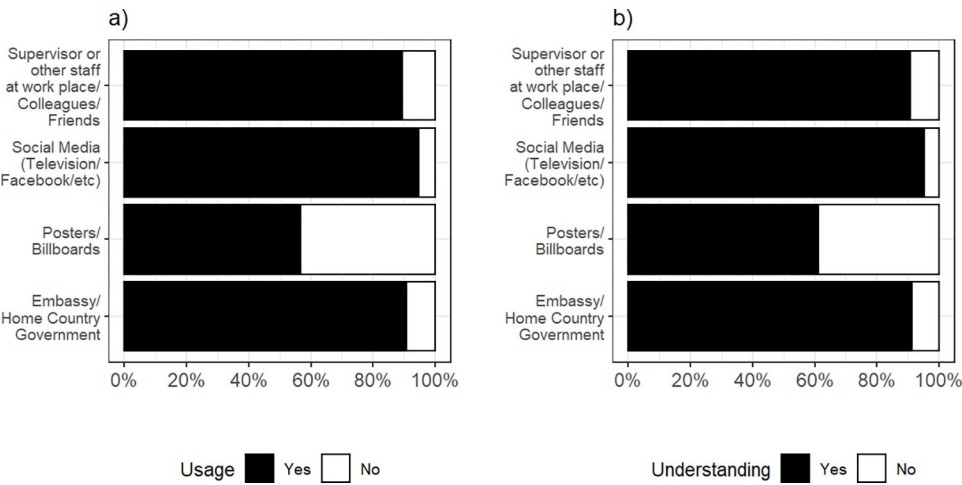

**Fig 1. Understanding and usage of DF information.** a) Percentage of the 403 respondents who obtained DF information from these sources. b) Percentage of the 403 respondents who understood DF information from these sources.

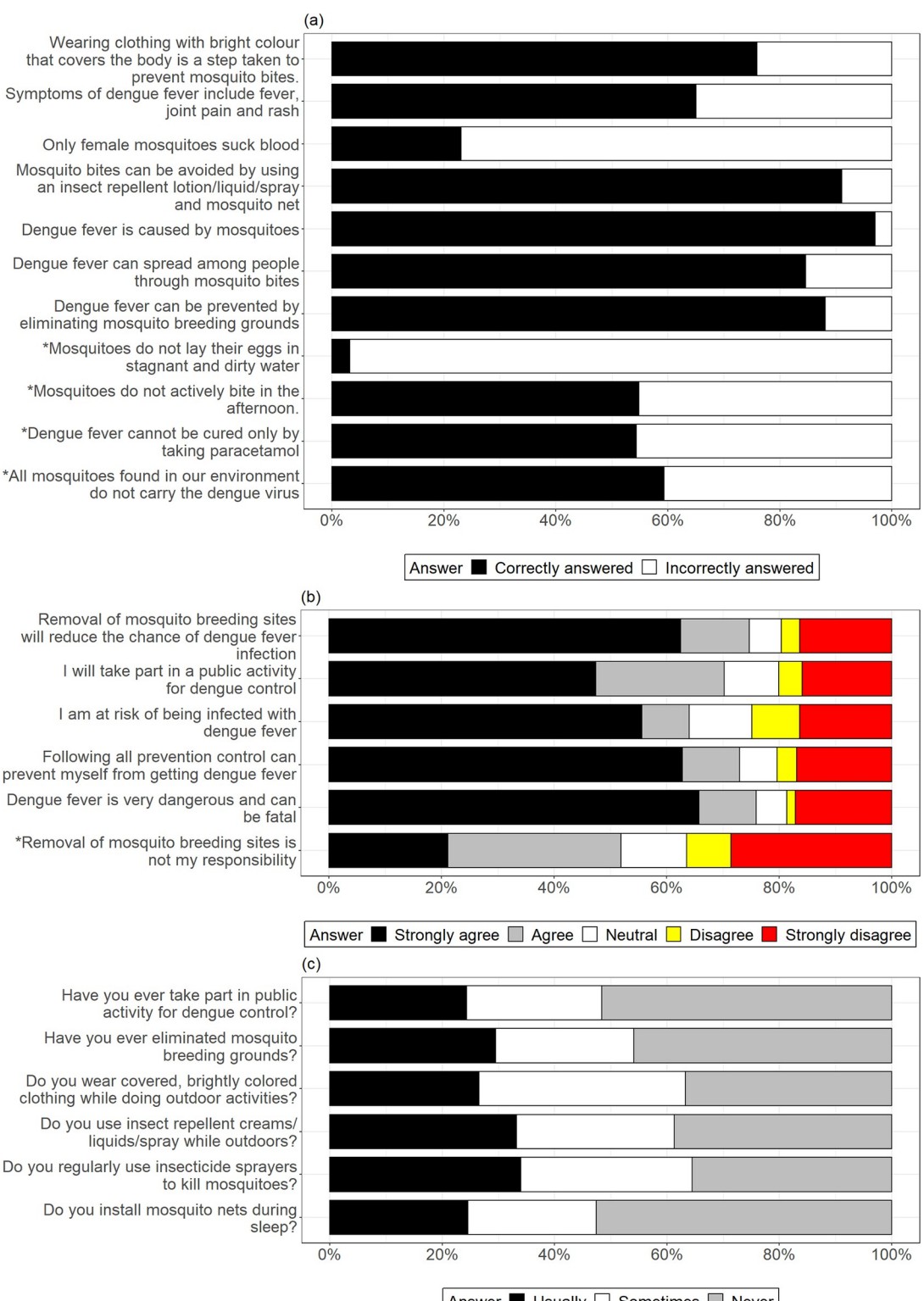

**Fig 2. Responses from 403 respondents for the dengue KAP statements.** a) Knowledge of DF Prevention. b) Attitudes Towards DF Prevention. c) Practices in DF Prevention. *These statements were originally negative but were reversely scored here and for the Linear Regression Modelling.

*insecticide sprayers to kill mosquitoes'*, with 64.5% usually/sometimes practising them. However, only 47.4% usually/sometimes *'installed mosquito nets during sleep'*.

## Identifying explanatory variables for KAP

The final PSEM model (i) based on the results of the linear mixed effects modelling (S1 Table and S1 Fig), (ii) with lowest AIC value (71.62), (iii) $p>0.05$ for all directed separation tests and Fisher's C test (C(30) = 31.62, $p$ = 0.386), and (iv) fitted the data well in post-hoc assessments (S2 Table) was retained and visualised (Fig 3).

'Knowledge' was negatively correlated with 'Practice' ($p$ = 0.033; S1J Fig) and positively correlated with 'Attitude' ($p<0.001$; S1K Fig). Higher level of 'Knowledge' was associated with better 'Understanding of DF Information from Social Media' ($p<0.001$; S1D Fig). 'Knowledge' was positively associated with 'Age' ($p$ = 0.009; S1A Fig), with respondents aged 30–39 years having higher knowledge than those aged <20 ($p$ = 0.001), 20–29 ($p$ = 0.005), 40–49 ($p<0.001$) and >49 ($p$ = 0.015). Those aged <20 had lower knowledge than those aged 20–29 ($p$ = 0.019) and 40–49 ($p$ = 0.042). 'Knowledge' was positively associated with 'Education' ($p$ = 0.032;

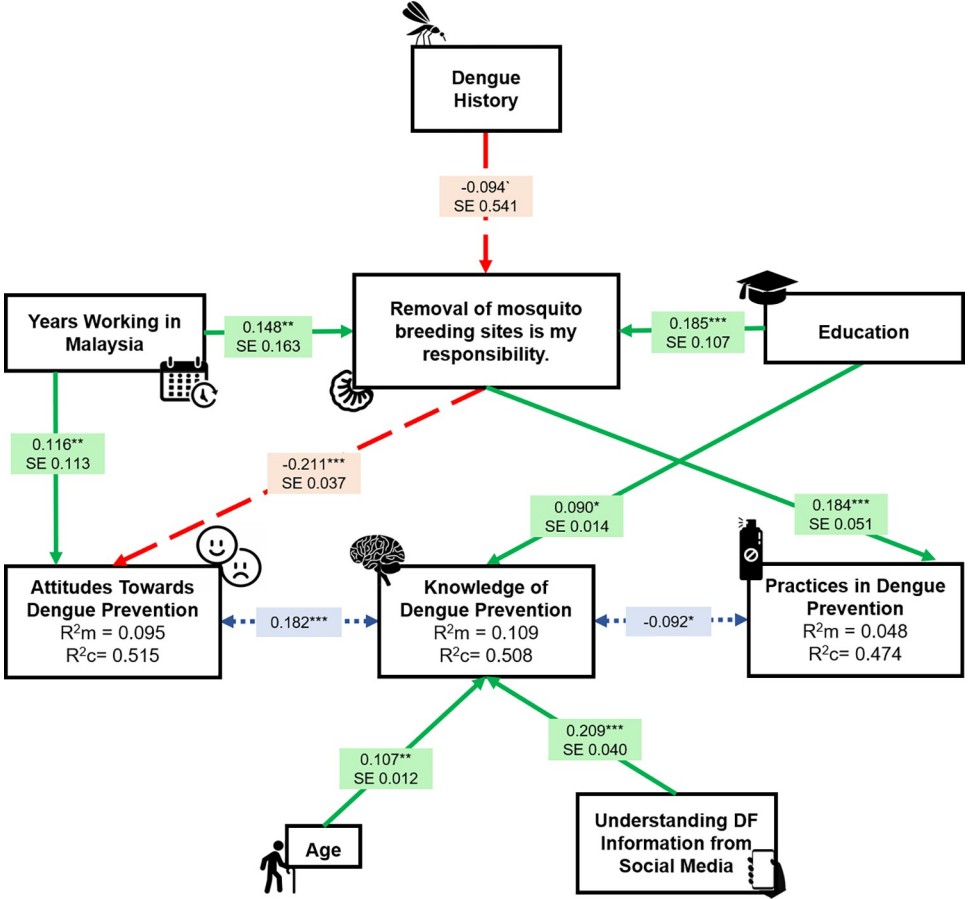

**Fig 3. Piecewise structural equation modelling for DF prevention KAP.** Arrowheads indicate the pathway of the relationship, where one variable influenced another. Green arrows with solid lines indicate a positive relationship between the two variables, red arrows with dashed lines indicate a negative relationship between the two variables whereas blue arrows with dotted lines indicate a correlation between the two variables. Note that the variables with binary responses were coded in such: Female/Yes = 1; Male/No = 0. Asterisks represent the significance levels of ps in increasing order: * = $p<0.05$, ** = $p<0.01$ and *** = $p<0.001$. $R^2m$ indicates marginal $R^2$, $R^2c$ indicates conditional $R^2$ whereas SE indicates standard error.

S1B Fig), with respondents who had completed high school having higher knowledge than those with no formal education ($p = 0.018$), those having completed primary school ($p = 0.002$) and university ($p = 0.007$). Those who had completed university had higher knowledge than those with primary schooling ($p < 0.001$).

'Attitude' was positively associated with 'Years Working in Malaysia' ($p < 0.030$; S1C Fig), with respondents who had worked in Malaysia for >20 years having higher attitudes than those having worked 10–20 years ($p = 0.043$). 'Attitude' was negatively associated with 'Responsibility' ($p < 0.001$; S1E Fig), with respondents who considered removal of mosquito breeding sites as their responsibility having lower attitudes.

'Practice' was positively associated with 'Responsibility' ($p < 0.001$; S1F Fig), with respondents who considered removal of mosquito breeding sites as their responsibility having higher practices. 'Responsibility' was associated with 'Dengue History' ($p = 0.056$; S1G Fig), with respondents who had previously suffered from DF strongly agreeing that removal of mosquito breeding sites is their responsibility. 'Responsibility' was positively associated with 'Education' ($p < 0.001$; S1I Fig), with respondents who had completed university having higher knowledge than those having completed primary school ($p < 0.001$). 'Responsibility' was positively associated with 'Years Working in Malaysia' ($p = 0.003$; S1H Fig), with respondents who had worked in Malaysia for <10 years having lower knowledge than those having worked 10–20 ($p < 0.020$) and >20 years ($p = 0.001$).

## Discussion

### Dengue prevention KAP

Overall, DF prevention knowledge (78.7%) and attitudes (78.6%) were generally high among our respondents, and were positively correlated, similar to other Malaysian studies [39–41]. Thus, possessing DF prevention knowledge leads to improved attitudes, which should ideally lead to behavioural change. However, few respondents considered the removal of mosquito breeding sites as their responsibility (36.5%), whereas only 47.4% strongly agreed to partake in public DF control activities (e.g., enacting a dengue control committee, removal of mosquito breeding sites). Active community involvement in DF control programs is integral to preventing DF; thus, health authorities should create programs that encourage public participation [17, 41].

Respondents seldom exercised DF preventive practices (56.5%). Knowledge and practices in DF prevention were negatively associated. Studies in Malaysia [40–43], Jamaica [44] and Thailand [45] also found that DF prevention knowledge does not necessarily translate into preventative practices. In migrant workers, this may be attributed to an inability to understand region-specific DF awareness campaigns due to limited proficiency in the local language [17, 46] and inability to afford precautionary measures (e.g., mosquito nets, insect repellent) due to low income [44]. Moreover, our study found that preventative measures were frequently practised by those who considered removing mosquito breeding sites as their own responsibility, who were, in turn, those with previous history of DF, similar to past studies [16, 47]. This suggests a lack of motivation to change current behaviour/habits, unless individuals experience and fully perceive the risks of DF [18, 41, 45, 48]. Therefore, we recommend these strategies for government agencies and non-governmental organisations (NGOs):

1. Provision of resources (e.g., insecticide, ovitrap) to remove monetary barriers to exercising good DF practices;

2. Interventions (e.g., educational material and training in multiple languages) to raise awareness and;

3. Programs to increase individuals' risk perception (e.g., emphasising the severity and complications of DF) to ensure that DF prevention knowledge is implemented.

## Targeted groups for training based on Socio-Demographic analyses

We found that respondents <30 years of age had the lowest knowledge scores whereas those aged >30 had higher knowledge scores, similar to past studies [16, 17]. This highlights the need to target younger individuals for increased DF education, especially since they are the most active working members of society and pose a greater risk of transmitting disease. In addition, those who had worked in Malaysia for >20 years had the highest knowledge scores, similar to studies in Malaysia [47], Indonesia [49] and Myanmar [50]. This is most likely because the longer respondents live in Malaysia, the more they know of the local culture [47], and the more proficient they become in the local language, allowing them to access information from health authorities [50].

Those who had completed at least high school showed high knowledge scores, similar to studies in Malaysia [17, 21, 40], Jamaica [44] and Thailand [45]. The more knowledgeable individuals are of DF, the more likely they are to implement preventative practices, provided that they fully understand the personal risk of contracting DF [19]. Health education interventions can increase the DF KAP of individuals, to ensure communal practice of preventative measures [16, 48]. Therefore, the Ministry of Health (MoH) should create DF education programs targeting the youth, different nationalities and newly arrived migrant workers to inculcate precautionary practices against DF.

## Social media as a tool for DF awareness

Among our respondents, 94.8% used social media to obtain information on DF similar to other studies [47, 48], and respondents' understanding of DF information from social media positively influenced their knowledge of DF. Similarly, a study found that 99.21% of respondents favoured an e-learning approach for DF health education, whereas the majority of respondents agreed that online educational resources were easy to understand, time-saving as they could be easily reviewed, and easy to share with others [51]. This, combined with the multiple benefits of social media, including wider public outreach, engagement with audiences and low cost [52], reiterates the potential of social media in raising DF awareness.

The Malaysian MoH already frequently shares infographics and animated clips on health-related topics in the Malay language, through their verified social media accounts. As of January 11, 2023, the MoH had 5.7M followers on Facebook (www.facebook.com/kementeriankesihatanmalaysia/), 2M on Twitter (www.twitter.com/kkmputrajaya), and 1.3M on Instagram (www.instagram.com/kementeriankesihatanmalaysia/). The higher follower count on Facebook suggests that the MoH's social media follower-base mostly consists of older individuals. The MoH's posts obtain the most likes on Instagram, indicating higher audience engagement on this platform. Thus, the MoH should optimise their content for platforms commonly used by the youth, such as TikTok (www.tiktok.com) and Instagram (www.instagram.com), by creating trendy reels and eye-catching stories. To target newly arrived migrant workers, with little to no knowledge of the local language, and those with no social media presence, the MoH could work with the respective embassies and employers of migrant workers to create DF awareness campaigns in their native languages.

Often, social media users are unable to differentiate between false and credible information [52]. Therefore, health authorities should assure social media users of the credibility of their educational material by providing reliable references in their educational material. The MoH

could also collaborate with NGOs that actively work to debunk false health information, such as Medical Mythbusters Malaysia (https://www.facebook.com/MedicalMythbustersMalaysia/) to cultivate critical thinking among social media users. Another study observed that 66.1% of respondents felt that DF health information delivered via social media was inadequate [48]. Thus, social media must be used as a complement to traditional methods (e.g., physical brochures, lectures), rather than a replacement. Moreover, deeply ingrained habits cannot be reformed overnight, suggesting the need for intervention strategies that inculcate life-long learning; frequent reminders via social media would foster such a positive feedback loop between DF KAP [41, 48, 51].

## Conclusion

This study has revealed the need for increasing community participation in DF control, through tailored interventions and social media awareness campaigns, to break deeply ingrained habits and foster preventative practices. Overall, these findings can aid respective embassies and the Malaysian government in developing targeted health interventions for migrant workers, to protect their own health and minimise disease transmission.

## Supporting information

**S1 Table. Final regression models.** The final models were selected by removing predictors from a global model sequentially until all predictors in the model met the preselected criterion ($p<0.05$) and all those outside did not. The marginal $R^2$ value reflects the variance explained by fixed factors, whereas the conditional $R^2$ represents the variance explained by both fixed and random factors [32].
(DOCX)

**S2 Table. Post hoc analysis of PSEM model.** The Tucker–Lewis index (TLI) allows for comparisons between the proposed and null model whilst comparative fit index (CFI), more specifically, measures the improvement in non-centrality between the two models, with values>0.9 indicating good model fit [37]. The standardised root mean square residual (SRMR) assesses the extent to which the sample variance-covariance data fits the PSEM, with values<0.05 providing substantial support for the model [37].
(DOCX)

**S1 Fig. Descriptive analyses based on the three linear regression models constructed from 403 responses.** (a) 'Practice' plotted against 'Knowledge', (b) 'Attitude' plotted against 'Knowledge', (c) 'Practice plotted against 'Attitude', (d) 'Knowledge' plotted against 'Age', (e) 'Knowledge' plotted against 'Gender', (f) 'Knowledge' plotted against 'Years Working in Malaysia', (g) 'Knowledge' plotted against 'Understanding DF Information from Social Media', (h) 'Attitude' plotted against 'Education', (i) 'Attitude' plotted against 'Gender', (j) 'Practice' plotted against 'Gender', (k) 'Practice' plotted against 'Understanding DF Information from Social Media'. Line of best fit (blue) with 90% confidence interval (orange) was plotted for (a), (b) and (c). Responses were superimposed on predicted group mean (grey) and standard error of means (blue bar) for (d), (e), (f), (g), (h), (i), (j) and (k).
(TIF)

**S1 File. Dengue fever prevention knowledge, attitude and practice questionnaire.**
(ZIP)

**S2 File. Dengue fever prevention knowledge, attitude and practice survey raw data (403 respondents).**
(ZIP)

**S3 File.**
(PDF)

**S4 File.**
(PDF)

**S1 Data.**
(XLSX)

## Acknowledgments

We express our sincere thanks to the coordinators; Mr. Ramesh Kumar Pajiyar and Ms. Babie de Vera for helping us during the data collection, and to all volunteers that participated in the study.

## Author Contributions

**Conceptualization:** Adzzie Shazleen Azman, Norhidayu Sahimin.

**Data curation:** Maryam N. Chaudhary.

**Formal analysis:** Maryam N. Chaudhary, Voon-Ching Lim.

**Funding acquisition:** Adzzie Shazleen Azman, Norhidayu Sahimin.

**Investigation:** Adzzie Shazleen Azman, Norhidayu Sahimin.

**Methodology:** Maryam N. Chaudhary, Voon-Ching Lim, Adzzie Shazleen Azman, Norhidayu Sahimin.

**Project administration:** Norhidayu Sahimin.

**Resources:** Adzzie Shazleen Azman, Norhidayu Sahimin.

**Supervision:** Voon-Ching Lim, Adzzie Shazleen Azman.

**Validation:** Maryam N. Chaudhary, Voon-Ching Lim, Erwin Martinez Faller, Pramod Regmi, Nirmal Aryal, Siti Nursheena Mohd Zain, Adzzie Shazleen Azman, Norhidayu Sahimin.

**Visualization:** Maryam N. Chaudhary.

**Writing – original draft:** Maryam N. Chaudhary, Voon-Ching Lim, Adzzie Shazleen Azman.

**Writing – review & editing:** Erwin Martinez Faller, Pramod Regmi, Nirmal Aryal, Siti Nursheena Mohd Zain, Norhidayu Sahimin.

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
