## [Decision Letter · Decision Letter 0]

2 Jul 2023

PONE-D-23-14903Assessing the Basic Knowledge and Awareness of Dengue Fever Prevention among Migrant Workers in Klang Valley, MalaysiaPLOS ONE

Dear Dr. Sahimin,

Thank you for submitting your manuscript to PLOS ONE. After careful consideration, we feel that it has merit but does not fully meet PLOS ONE’s publication criteria as it currently stands. Therefore, we invite you to submit a revised version of the manuscript that addresses the points raised during the review process.

We look forward to receiving your revised manuscript.

Kind regards,

Ian Christopher N Rocha, MD, MBA, MHSS

Academic Editor

PLOS ONE

Additional Editor Comments:

Kindly address the comments of the reviewers.

Reviewers' comments:

Reviewer's Responses to Questions

**Comments to the Author**

1. Is the manuscript technically sound, and do the data support the conclusions?

Reviewer #1: Yes

Reviewer #2: Yes

2. Has the statistical analysis been performed appropriately and rigorously? 

Reviewer #1: Yes

Reviewer #2: Yes

3. Have the authors made all data underlying the findings in their manuscript fully available?

Reviewer #1: Yes

Reviewer #2: No

4. Is the manuscript presented in an intelligible fashion and written in standard English?

Reviewer #1: Yes

Reviewer #2: Yes

5. Review Comments to the Author

Reviewer #1: I read the research article entitled “Assessing the Basic Knowledge and Awareness of Dengue Fever Prevention among Migrant Workers in Klang Valley, Malaysia”

This conducted on immigrant workers in Malaysia, to detect their knowledge, attitude, and preventive measures about dengue fever infection.

The results show that the participant knowledge was positively correlated with attitude but negatively with practices. Older participant, who had completed higher education, obtained higher knowledge scores.

Those with working experience of >20 years in Malaysia obtained higher attitude scores.

This survey study is important to detect the knowledge gap and how to improve the practice, awareness toward dengue fever infection which can lead to individual’s preventive measure to avoid dengue infection.

The paper is good design; good work but need some improvement as

Major issues:

Methods section:

1. “A pilot study involving five Indonesians”, the pilot study should run at least on 30 participants not 5.

2. The internal consistency of the KAP questions/statements in the questionnaire was assessed using Cronbach’s Alpha test, this test mostly used in pilot sample which contain at least 30 participant, if the result more than 0.7, then run the questionnaire to all participant in the study.

3. There are multiple details and repeated data in data analysis. Please revise it and remove unnecessary data.

Minor issues:

Introduction:

1. You write: Dengue fever is fatal. Although Approximately 10% of dengue cases progress to severe disease, and the majority of cases have mild disease.

2. Following the manufacturing industry (36%), most migrant workers in Malaysia are employed in the construction industry (19%). This sentence has something wrong, please revise it.

Results section:

1. Write the table or the figure numbers at the end of the sentences.

2. Multiple repeated the name of the test in the results section is not advisable as for example ; Dunn’s test , please remove it.

Reference section: please revise it

1. References N 3, 8, 10, 23, 25, 26, 33 from website, please write it correctly with access day, month, year, and URL link.

2. Refrences N. 7, 11, 12, 14, 15, 21,39, 43 please write the pages numbers.

You need English editing, revision and good writing of methods and results section

Reviewer #2: Summary

In this study, the authors arguen that Among the global cases of Dengue Fever (DF) migrant workers in Malaysia are a significant risk group due to exposure to mosquito breeding sites and low health literacy and that, future Preventive interventions to DF require an assessment of the Knowledge, Attitudes and preventive practices (KAP) of the migrant workers in Malaysia.

The authors conducted a survey with 403 Nepali, Filipino and Indonesian migrant workersthrough phone interviews and online self-administered questionnaires, and used Piecewise structural equation modelling to identify predictor variables for DF KAP.

Respondents’ knowledge was positively correlated with attitude but negatively with practices. Older respondents, who had completed higher education, obtained higher knowledge scores. Similarly, those with working experience of >20 years in Malaysia obtained higher attitude scores. Respondents with a previous history of DF strongly considered the removal of mosquito breeding sites as their own responsibility, hence tended to frequently practise DF preventive measures. Respondents’ knowledge was also positively correlated to their understanding of DF information sourced from social media platforms.

Impact and significance of the study

The Impact of this study is that for the first time, this kind of study was carried out among Migrant workers in Malaysia, and that his knowledge could inform future interventions.

Quality of the work

The manuscript was written in standard english that is easy to read and understand. The statistical analysis methods used in this study are adquate, however, the data collection and data analysis requires more explanation, as it was not easy to follow what was being done.I have listed below some of my concerns and questions.

Concerns and Questions

-The manuscript was submitted without numbering the lines of the text, which makes it difficult to reference them in the review.

- In the first sentence of the Abstract, please indicate whether "390 million Dengue virus infetions" is a global or local prevalence. It is indicated in the introduction section, but please do the same for the abstract.

- In the line 6 of the last paragraph of introduction, "(i) better understand the level of and relationship between KAP" - Maybe it is better to not abbreviate KAP here but rather write it in full. It makes it easier to understand the sentence.

- Why did you choose to use 455 study participants? please explain the reason for choosing this sample size. Was it based on some sample size estimation method?

-Please explain the Inclusion and Exclusion criteria for selecting participants.

- Please explain the nature of the variables used in the analysis. Were the answers converted to some number scale? are these categorical variables? What is the range of their values. This will help readers understand the reason for your choice of statistical analytical methods.

-In the selection of explanatory variables for analysis, please explain which variables were initially considered, how and why you eliminated some variables and what were the final variables included in the model. You provided very little explanation of the analysis.

6. PLOS authors have the option to publish the peer review history of their article (what does this mean?). If published, this will include your full peer review and any attached files.

Reviewer #1: No

Reviewer #2: **Yes: **Daniel Addo-Gyan

---

## [Author Response · Author response to Decision Letter 0]

14 Jul 2023

Dear Academic Editor and Reviewers,

Thank you for your comments; we appreciate the time and effort taken to review our manuscript. Please see our responses to your comments below.

General Comments:

Manuscript has been revised to meet PLOS ONE’s style requirements.

2. We note that you have indicated that data from this study are available upon request. PLOS only allows data to be available upon request if there are legal or ethical restrictions on sharing data publicly. If there are no ethical or legal restrictions, please upload the minimal anonymized data set necessary to replicate your study findings as either Supporting Information files or to a stable, public repository and provide us with the relevant URLs, DOIs, or accession numbers. For a list of acceptable repositories, please see http://journals.plos.org/plosone/s/data-availability#loc-recommended-repositories.

Data has been made available as Supplementary File 2.

Reviewer #1

3. “A pilot study involving five Indonesians”, the pilot study should run at least on 30 participants not 5. ‘The internal consistency of the KAP questions/statements in the questionnaire was assessed using Cronbach’s Alpha test’, this test mostly used in pilot sample which contain at least 30 participant, if the result more than 0.7, then run the questionnaire to all participant in the study.

The following sentence has been modified for clarity (Page 4, Line 91): ‘A pilot study was conducted to test the comprehension of the questionnaire wherein five study leaders distributed the questionnaire to ~10 respondents each.’

A Cronbach’s alpha value of ≥0.7 was used to indicate acceptable internal consistency of the questions to represent a single construct. Statements were removed from or retained in the final construct on this basis. We followed the methodology of Reference 24.

4. There are multiple details and repeated data in data analysis. Please revise it and remove unnecessary data.

The manuscript has been edited accordingly.

5. You write: Dengue fever is fatal. Although Approximately 10% of dengue cases progress to severe disease, and the majority of cases have mild disease.

Sentence has been rephrased to (Page 3, Line 57) ‘DF can be fatal, with symptoms including high fever, body pains and persistent vomiting’.

6. Following the manufacturing industry (36%), most migrant workers in Malaysia are employed in the construction industry (19%). This sentence has something wrong, please revise it.

Sentence has been rephrased to (Page 3, Line 71) ‘Most migrant workers in Malaysia are employed in the manufacturing (36%) and construction industry (19%)’.

7. Write the table or the figure numbers at the end of the sentences.

Table and figure numbers have been listed at the end of each sentence. There are a few exceptions to ensure the reader understands which figure/table to refer to for each component of the sentence.

E.g., ‘The final PSEM model (i) based on the results of the linear mixed effects modelling (S1 Table and S1 Fig), (ii) with lowest AIC value (71.62), (iii) p>0.05 for all directed separation tests and Fisher’s C test (C(30)=31.62, p=0.386), and (iv) fitted the data well in post-hoc assessments (S2 Table) was retained and visualised (Fig 3).’

8. Multiple repeated the name of the test in the results section is not advisable as for example ; Dunn’s test , please remove it.

The manuscript has been edited accordingly.

9. References N 3, 8, 10, 23, 25, 26, 33 from website, please write it correctly with access day, month, year, and URL link.

All references have been updated accordingly.

10. Refrences N. 7, 11, 12, 14, 15, 21,39, 43 please write the pages numbers.

All references have been updated accordingly.

Reviewer #2

11. The manuscript was submitted without numbering the lines of the text, which makes it difficult to reference them in the review.

The manuscript has been edited accordingly.

12. In the first sentence of the Abstract, please indicate whether "390 million Dengue virus infetions" is a global or local prevalence. It is indicated in the introduction section, but please do the same for the abstract.

Sentence has been rephrased to (Page 2, Line 34) ‘Globally, 390 million dengue virus infections occur per year’.

13. In the line 6 of the last paragraph of introduction, "(i) better understand the level of and relationship between KAP" - Maybe it is better to not abbreviate KAP here but rather write it in full. It makes it easier to understand the sentence.

The manuscript has been edited accordingly.

14. Why did you choose to use 455 study participants? please explain the reason for choosing this sample size. Was it based on some sample size estimation method?

The following sentence has been added for clarity (Page 5, Line 120): ‘The Krejcie and Morgan formula was applied, with a 5% margin of error and 95% confidence interval, resulting in a minimum sample size of 385 participants.’

15. Please explain the Inclusion and Exclusion criteria for selecting participants.

The Inclusion and Exclusion criteria was stated as follows (Page 5, Line 114): ‘Respondents from the Philippines and Nepal were invited for this study due to past collaboration with their respective coordinators, whereas Indonesian respondents were invited as they constitute the majority of migrant workers in Malaysia’. 

16. Please explain the nature of the variables used in the analysis. Were the answers converted to some number scale? are these categorical variables? What is the range of their values. This will help readers understand the reason for your choice of statistical analytical methods.

The following sentence has been added for clarity (Page 5, Line 144): ‘The categorical explanatory variables were converted to numeric binary (e.g. 0 = male and 1 = female) or ordinal (e.g., 1 = no formal education, 2 = primary school, 3 = high school, 4 = university)’.

17. In the selection of explanatory variables for analysis, please explain which variables were initially considered, how and why you eliminated some variables and what were the final variables included in the model. You provided very little explanation of the analysis.

Explanatory variables that were initially considered were stated as follows (Page 5, Line 140): ‘The explanatory variables were ‘Gender’, ‘Age’, ‘Years Working in Malaysia’, ‘Education’, ‘Occupation’, ‘Dengue History’, ‘Understanding of DF Information from Social Media’ and ‘Responsibility’’.

Basis for variable elimination was stated as follows (Page 6, Line 146): ‘The final models were selected by removing predictors from a global model sequentially until all predictors in the model met the preselected criterion (p<0.05) and all those outside did not’.

The final variables included in the model are discussed in the Results (Pages 8-9 ) and Discussion sections. The model with the final selected variables is visualised in Figure 3.

---

## [Decision Letter · Decision Letter 1]

10 Nov 2023

PONE-D-23-14903R1Assessing the basic knowledge and awareness of dengue fever prevention among migrant workers in Klang Valley, MalaysiaPLOS ONE

Dear Dr. Sahimin,

Thank you for submitting your manuscript to PLOS ONE. After careful consideration, we feel that it has merit but does not fully meet PLOS ONE’s publication criteria as it currently stands. Therefore, we invite you to submit a revised version of the manuscript that addresses the points raised during the review process.

ACADEMIC EDITOR: Kindly address the comments of Reviewer #3.==============================

We look forward to receiving your revised manuscript.

Kind regards,

Ian Christopher N Rocha, MD, MBA, MHSS

Academic Editor

PLOS ONE

Journal Requirements:

Additional Editor Comments: Kindly address the comments of Reviewer #3.

Reviewers' comments:

Reviewer's Responses to Questions

**Comments to the Author**

1. If the authors have adequately addressed your comments raised in a previous round of review and you feel that this manuscript is now acceptable for publication, you may indicate that here to bypass the “Comments to the Author” section, enter your conflict of interest statement in the “Confidential to Editor” section, and submit your "Accept" recommendation.

Reviewer #1: All comments have been addressed

Reviewer #2: All comments have been addressed

Reviewer #3: (No Response)

2. Is the manuscript technically sound, and do the data support the conclusions?

Reviewer #1: Yes

Reviewer #2: Yes

Reviewer #3: Partly

3. Has the statistical analysis been performed appropriately and rigorously? 

Reviewer #1: Yes

Reviewer #2: Yes

Reviewer #3: Yes

4. Have the authors made all data underlying the findings in their manuscript fully available?

Reviewer #1: Yes

Reviewer #2: No

Reviewer #3: No

5. Is the manuscript presented in an intelligible fashion and written in standard English?

Reviewer #1: Yes

Reviewer #2: Yes

Reviewer #3: Yes

6. Review Comments to the Author

Reviewer #1: The research article entitled “Assessing the Basic Knowledge and Awareness of Dengue Fever Prevention among Migrant Workers in Klang Valley, Malaysia”

This was conducted on immigrant workers in Malaysia, to detect their knowledge, attitude, and preventive measures about dengue fever infection.

The results show that the participant knowledge was positively correlated with attitude but negatively with practices. Older participants, who had completed higher education, obtained higher knowledge scores.

Those with working experience of >20 years in Malaysia obtained higher attitude scores.

This survey study is important to detect the knowledge gap and how to improve the practice, and awareness toward dengue fever infection which can lead to individual preventive measures to avoid dengue infection.

1. The comments needed were corrected in the new article

Reviewer #2: (No Response)

Reviewer #3: Thank you for the revision you have added into this manuscript. The manuscript discusses the KAP of migrant workers on dengue fever, along with its prevention and measurements. The article is concise with novel knowledge and information on the topic. However, there are several things I would like to address:

1. Is there any validity test on different versions of questionnaire used in the data collection, as you mention particularly Indonesian workers used Indonesian instead of English. Will this become a potential bias in methodology? Please elaborate your answer, and if there is a risk of bias or skewness arise, also add your reasonings in including the data.

2. I would suggest for you to include individual analysis of each factor on KAP, instead of the final model stated in the results section. As also mentioned by the previous reviewer concerning the lack of details in selecting factors for final model, I believe your explanation can be added instead to the text.

3. On discussion, please add further details and discussion on the results, especially the correlation between KAP and previously mentioned factors. I am personally intrigued by the negative correlation of knowledge and practice. Do you have any explanation or possible reasoning for such things?

Overall, I believe the manuscript is well-written. I hope you can answer some questions of mine.

7. PLOS authors have the option to publish the peer review history of their article (what does this mean?). If published, this will include your full peer review and any attached files.

Reviewer #1: No

Reviewer #2: **Yes: **Daniel Kweku Addo-Gyan

Reviewer #3: **Yes: **Lowilius Wiyono

---

## [Author Response · Author response to Decision Letter 1]

15 Nov 2023

Dear Academic Editor and Reviewers,

Thank you for your comments; we appreciate the time and effort taken to review our manuscript. Please see our responses to your comments below.

General Comments:

All cited references are correct and still available online, to the best of our knowledge. 

Reviewer #3

2. Is there any validity test on different versions of questionnaire used in the data collection, as you mention particularly Indonesian workers used Indonesian instead of English. Will this become a potential bias in methodology? Please elaborate your answer, and if there is a risk of bias or skewness arise, also add your reasonings in including the data.

Tests for validity are discussed in detail as follows (Page 5, Line 126): ‘The internal consistency of the KAP questions/statements in the questionnaire was assessed using Cronbach’s Alpha test, using ‘alpha()’ from the ‘psych’ package 26. A Cronbach’s alpha value of ≥0.7 indicates acceptable internal consistency of the questions to represent a single construct 27-29.’

Data collection differed between the three nationalities (Page 5, Line 116): ‘Filipino respondents completed the questionnaire in English. Due to limited English proficiency among Nepali and Indonesian respondents, Nepali respondents underwent a phone interview conducted by their coordinator in their native language, whereas Indonesian respondents completed the questionnaire in the Indonesian language.’

To mitigate the effects of any potential bias, the following measure was taken (Page 5, Line 143): ‘Nationality’ was coded as a random error as the method of response collection varied between nationalities (e.g., English/Indonesian questionnaire, and phone interview).

3. I would suggest for you to include individual analysis of each factor on KAP, instead of the final model stated in the results section. As also mentioned by the previous reviewer concerning the lack of details in selecting factors for final model, I believe your explanation can be added instead to the text.

The individual linear regression models for each of knowledge, attitude and practices are included in the Appendix (S1 Table). The final, piecewise structural equation model was based on the aforementioned individual regression analyses. Piecewise structural equation modelling can accurately gauge complex multivariate relationships, whilst allowing variables to serve as both predictors and responses.

Details for selecting factors are included in the text as follows:

● Explanatory variables that were initially considered (Page 5, Line 140): ‘The explanatory variables were ‘Gender’, ‘Age’, ‘Years Working in Malaysia’, ‘Education’, ‘Occupation’, ‘Dengue History’, ‘Understanding of DF Information from Social Media’ and ‘Responsibility’’.

● Basis for variable elimination (Page 6, Line 146): ‘The final models were selected by removing predictors from a global model sequentially until all predictors in the model met the preselected criterion (p<0.05) and all those outside did not’.

4. On discussion, please add further details and discussion on the results, especially the correlation between KAP and previously mentioned factors. I am personally intrigued by the negative correlation of knowledge and practice. Do you have any explanation or possible reasoning for such things?

Each relationship depicted in Figure 3 is explored in the discussion section as follows:

● Correlation between knowledge, attitudes and practices under ‘Dengue Prevention KAP’ (Page 9, Line 236)

● Correlation between KAP and sociodemographic variables under ‘Targeted Groups for Training Based on Socio-Demographic Analyses’ (Page 10, Line 263)

● Correlation between KAP and social media under ‘Social Media as a Tool for DF Awareness’ (Page 11, 281)

The negative correlation between knowledge and practice was explored as follows (Page 10, Line 245): 

‘Respondents seldom exercised DF preventive practices (56.5%). Knowledge and practices in DF prevention were negatively associated. Studies in Malaysia 40-43, Jamaica 44 and Thailand 45 also found that DF prevention knowledge does not necessarily translate into preventative practices. In migrant workers, this may be attributed to an inability to understand region-specific DF awareness campaigns due to limited proficiency in the local language 17, 46 and inability to afford precautionary measures (e.g., mosquito nets, insect repellent) due to low income 44. Moreover, our study found that preventative measures were frequently practised by those who considered removing mosquito breeding sites as their own responsibility, who were, in turn, those with previous history of DF, similar to past studies 16 47. This suggests a lack of motivation to change current behaviour/habits, unless individuals experience and fully perceive the risks of DF 18, 41, 45, 48.

---

## [Decision Letter · Decision Letter 2]

8 Jan 2024

Assessing the basic knowledge and awareness of dengue fever prevention among migrant workers in Klang Valley, Malaysia

PONE-D-23-14903R2

Dear Dr. Sahimin,

We’re pleased to inform you that your manuscript has been judged scientifically suitable for publication and will be formally accepted for publication once it meets all outstanding technical requirements.

Kind regards,

Harapan Harapan, MD, PhD

Academic Editor

PLOS ONE

Additional Editor Comments (optional):

Reviewers' comments:

Reviewer's Responses to Questions

**Comments to the Author**

1. If the authors have adequately addressed your comments raised in a previous round of review and you feel that this manuscript is now acceptable for publication, you may indicate that here to bypass the “Comments to the Author” section, enter your conflict of interest statement in the “Confidential to Editor” section, and submit your "Accept" recommendation.

Reviewer #1: (No Response)

Reviewer #2: All comments have been addressed

2. Is the manuscript technically sound, and do the data support the conclusions?

Reviewer #1: Yes

Reviewer #2: Yes

3. Has the statistical analysis been performed appropriately and rigorously? 

Reviewer #1: Yes

Reviewer #2: Yes

4. Have the authors made all data underlying the findings in their manuscript fully available?

Reviewer #1: Yes

Reviewer #2: No

5. Is the manuscript presented in an intelligible fashion and written in standard English?

Reviewer #1: Yes

Reviewer #2: Yes

6. Review Comments to the Author

Reviewer #1: The research article entitled “Assessing the Basic Knowledge and Awareness of Dengue Fever Prevention among Migrant Workers in Klang Valley, Malaysia”

This conducted on immigrant workers in Malaysia, to detect their knowledge, attitude, and preventive measures about dengue fever infection.

The results show that the participant knowledge was positively correlated with attitude but negatively with practices. Older participant, who had completed higher education, obtained higher knowledge scores.

Those with working experience of >20 years in Malaysia obtained higher attitude scores.

This survey study is important to detect the knowledge gap and how to improve the practice, awareness toward dengue fever infection which can lead to individual’s preventive measure to avoid dengue infection.

1. The article is good written and scientific sound, I have no comments

Reviewer #2: (No Response)

7. PLOS authors have the option to publish the peer review history of their article (what does this mean?). If published, this will include your full peer review and any attached files.

Reviewer #1: No

Reviewer #2: **Yes: **Daniel Addo-Gyan

---

## [Editor Report · Acceptance letter]

24 Jan 2024

PONE-D-23-14903R2 

PLOS ONE

Dear Dr. Sahimin, 

I'm pleased to inform you that your manuscript has been deemed suitable for publication in PLOS ONE. Congratulations! Your manuscript is now being handed over to our production team.

Kind regards, 

on behalf of

Dr. Harapan Harapan 

Academic Editor

PLOS ONE